# Chance-Constrained POMDP Planning with Learned Neural Network Surrogates

**Robert J. Moss**[1], **Arec Jamgochian**[1], **Johannes Fischer**[1,2],
**Anthony Corso**[1], and **Mykel J. Kochenderfer**[1]

[1]Stanford University, Stanford, CA
[2]Karlsruhe Institute of Technology (KIT), Karlsruhe, Germany

{mossr, arec, acorso, mykel}@stanford.edu, johannes.fischer@kit.edu

## Abstract

To plan safely in uncertain environments, agents must balance utility with safety constraints. Safe planning problems can be modeled as a chance-constrained partially observable Markov decision process (CC-POMDP) and solutions often use expensive rollouts or heuristics to estimate the optimal value and action-selection policy. This work introduces the *ConstrainedZero* policy iteration algorithm that solves CC-POMDPs in belief space by learning neural network approximations of the optimal value and policy with an additional network head that estimates the failure probability given a belief. This failure probability guides safe action selection during online Monte Carlo tree search (MCTS). To avoid overemphasizing search based on the failure estimates, we introduce $\Delta$-MCTS, which uses adaptive conformal inference to update the failure threshold during planning. The approach is tested on a safety-critical POMDP benchmark, an aircraft collision avoidance system, and the sustainability problem of safe $CO_2$ storage. Results show that by separating safety constraints from the objective we can achieve a target level of safety without optimizing the balance between rewards and costs.

## 1 Introduction

When developing safety-critical agents to make sequential decisions in uncertain environments, planning and reinforcement learning algorithms often formulate the problem as a partially observable Markov decision process (POMDP) with the objective of maximizing a scalar-valued reward function [Kochenderfer *et al.*, 2022]. To ensure adequate safety, the scalar reward is tuned to balance the goals of the agent while penalizing undesired behavior or failures. Recently, chance-constrained POMDPs (CC-POMDPs) have been used to frame the safe planning problem by separating the reward function into a constrained problem [Santana *et al.*, 2016]. The objective of CC-POMDPs is to maximize rewards while satisfying safety constraints. Lauri *et al.* [2022] highlight the limitations of such chance-constrained POMDP algorithms and the need for scalable approaches to solve large-scale, long-horizon CC-POMDPs in practice.

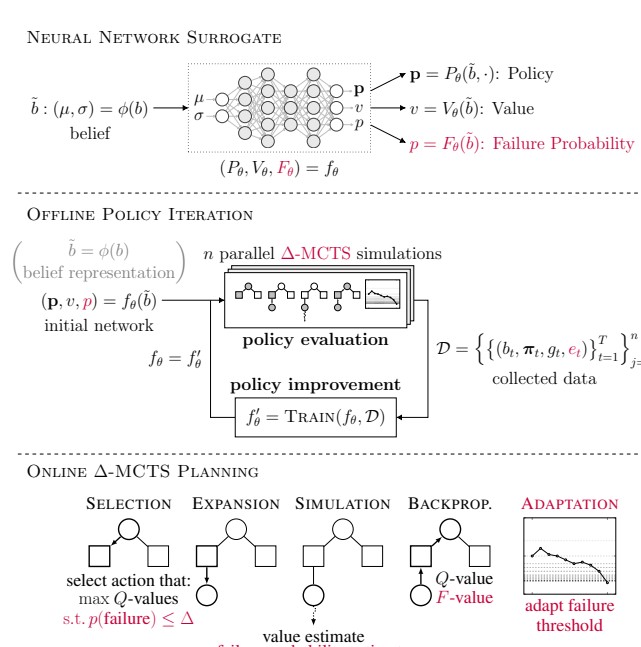

Figure 1: Elements of *ConstrainedZero* for CC-POMDP planning.

To address scalability and applicability to continuous state and observation spaces, we introduce the *ConstrainedZero* policy iteration algorithm that combines offline neural network training of the value function, the action-selection policy, and the failure probability predictor with online Monte Carlo tree search (MCTS) to improve the policy through planning. ConstrainedZero is a direct extension to the POMDP belief-state planning algorithm BetaZero [Moss *et al.*, 2024a] and the family of AlphaZero algorithms [Silver *et al.*, 2018], with extensions shown in red in fig. 1. Along with an open-source implementation,[1] our main contributions are 1) we introduce $\Delta$-MCTS, an anytime algorithm for MDPs (applied to belief-state MDPs) that estimates failure probabilities along with $Q$-values and adjusts the failure probability threshold using adaptive conformal inference [Gibbs and Candes, 2021], and 2) we introduce ConstrainedZero, a policy iteration algorithm that extends BetaZero for CC-POMDPs,

---

[1]https://github.com/sisl/ConstrainedZero.jl

which includes an additional network head that estimates failure probability given a belief and uses $\Delta$-MCTS with the neural network surrogate to prioritize promising safe actions, replacing expensive rollouts or domain-specific heuristics. See Moss *et al.* [2024b] for the full ConstrainedZero paper.

## 2 Problem Formulation

This section formulates the safe planning problem we studied.

**POMDPs.** The partially observable Markov decision process (POMDP) is a framework for sequential decision making problems where the agent has uncertainty over their state in the environment [Kochenderfer *et al.*, 2022]. The POMDP consists of a state space $\mathcal{S}$, an action space $\mathcal{A}$, an observation space $\mathcal{O}$, a transition model $T$, a reward model $R$, an observation model $O$, and a discount factor $\gamma \in [0, 1]$. When solving POMDPs, the objective is to find a policy $\pi(b)$ given a belief $b$ over the unobserved state and return an action $a \in \mathcal{A}$ that maximizes the *value* of the belief, which is the expected discounted sum of rewards when following the policy $\pi$.

**Belief-state MDPs.** Every POMDP can be cast as an MDP by simply treating the belief as the state. In doing so, one can construct a belief-state MDP (BMDP) with the belief space $\mathcal{B}$ of the original POMDP as the MDP state space, while using the same action space $\mathcal{A}$, the belief-based reward model $R_b$, and a transition function $b' \sim T_b(\cdot \mid b, a)$ to get an updated belief $b'$. The belief update may be done exactly or using approximations such as a Kalman filter [Wan and Van Der Merwe, 2000] or particle filter [Thrun *et al.*, 2005].

**Chance-constrained planning.** When dealing with safety-critical sequential decision making problems, separating safety constraints from the objective allows for solvers to target an adequate level of safety while simultaneously maximizing rewards. This is in contrast to designing a single reward function to balance the rewards from the goals and penalties from violating safety. The chance-constrained POMDP (CC-POMDP) defines a failure set $\mathcal{F}$ that includes all state-action pairs $(s, a) \in \mathcal{S} \times \mathcal{A}$ that fail and a bound $\Delta \in [0, 1]$ on the probability, or chance, of a failure event occurring. Chance constraints are intuitive for users to define as they translate to the target failure probability of the agent, which is often the requirement for systems in industries such as aviation [Busch, 1985] and finance [Flannery, 1989]. The objective when solving CC-POMDPs is to maximize the value function while ensuring that the failure probability, or the chance constraint, is below the target threshold $\Delta$:

$$\underset{\pi}{\text{maximize}} \quad V^\pi(b_0) = \mathbb{E}_\pi \left[ \sum_{t=0}^\infty \gamma^t R_b(b_t, a_t) \mid b_0 \right] \quad (1)$$

$$\text{subject to} \quad F^\pi(b_0) = \mathbb{P}_\pi \left[ \bigvee_{t=0}^\infty \left( (s_t, a_t) \in \mathcal{F} \right) \mid b_0 \right] \leq \Delta \quad (2)$$

The failure probability $F^\pi(b_t)$ is often called the *execution risk* of the policy $\pi$ computed from the belief $b_t$.

Therefore, the CC-POMDP is defined as the tuple $\langle \mathcal{S}, \mathcal{A}, \mathcal{O}, \mathcal{F}, T, R, O, \gamma, \Delta \rangle$. Our work casts the CC-POMDP to a chance-constrained belief-MDP (CC-BMDP). The CC-BMDP tuple $\langle \mathcal{B}, \mathcal{A}, F_b, T_b, R_b, \gamma, \Delta \rangle$ extends BMDPs with an immediate failure probability function $F_b : \mathcal{B} \times \mathcal{A} \to [0, 1]$

and a failure probability threshold $\Delta$. The immediate failure probability is computed using the failure set $\mathcal{F}$ as:

$$F_b(b, a) = \int_{s \in \mathcal{S}} b(s) \mathbb{1}\{(s, a) \in \mathcal{F}\} \, \mathrm{d}s \quad (3)$$

## 3 Approach

ConstrainedZero follows the BetaZero [Moss *et al.*, 2024a] policy iteration steps of *policy evaluation* and *policy improvement* while also collecting failure event indicators to train the failure probability network head. During policy evaluation, $n$ parallel $\Delta$-MCTS executions are run and a data set $\mathcal{D}$ is collected. The data set $\mathcal{D} = \left\{ \{b_t, \boldsymbol{\pi}_t, g_t, e_t\}_{t=1}^T \right\}_{j=1}^n$ is a tuple of the belief at episode time step $t$ denoted $b_t$, the tree policy $\boldsymbol{\pi}_t$, the return $g_t = \sum_{i=t}^T \gamma^{(i-t)} r_i$ based on the observed reward $r_i$ and discount factor $\gamma$, and the failure event indicator $e_t$, where $g_t$ and $e_t$ are computed at the end of the trajectory for all time $t \leq T$. The failure event is computed as the disjunction of all state and action pairs of the CC-POMDP in the execution trajectory to ensure that if a trajectory failed at some point the full trajectory is marked as a failure:

$$e_t = \mathbb{1}\left\{ \bigvee_{i=t}^T \left( (s_i, a_i) \in \mathcal{F} \right) \right\} \quad (4)$$

During policy improvement, the neural network is trained to minimize the MSE or MAE loss $\mathcal{L}_{V_\theta}(g_t, v_t)$ to regress the value function $v_t = V_\theta(\tilde{b}_t)$, minimize the cross-entropy loss $\mathcal{L}_{P_\theta}(\boldsymbol{\pi}_t, \mathbf{p}_t)$ to imitate the tree policy $\mathbf{p}_t = P_\theta(\tilde{b}_t)$, and additionally minimize the binary cross-entropy loss $\mathcal{L}_{F_\theta}(e_t, p_t)$ to regress the failure probability function $p_t = F_\theta(\tilde{b}_t)$, with added regularization using the $L_2$-norm of the weights $\theta$. The failure probability head of the neural network includes a final sigmoid layer to ensure the output can be interpreted as a probability in the range $[0, 1]$.

### 3.1 Adaptive Safety Constraints in $\Delta$-MCTS

When using online MCTS for CC-BMDP planning, two considerations have to be addressed: 1) how to estimate the observed failure probability in the tree search, and 2) how to select actions constrained by this failure probability.

At each belief-state and action node $(b, a)$, the immediate failure probability $p$ is computed using $p = F_b(b, a)$. An estimate of the future failure probability $p'$ can be computed using rollouts, which may be expensive, thus we use the trained neural network head for failure probability estimation $p' = F_\theta(\tilde{b}')$. Similar to the $Q$-value, we must compute the full failure probability of the trajectory from the immediate time step to the horizon, termed the $F$-value. The probability of a failure event $E$ between the current time $t$ and the horizon $T$ is given by $P(E_{t:T}) = p + (1 - p)p'$, assuming independence in the derivation. A discount $\delta$ is applied to control the influence of the future failure probability, resulting in $p = p + \delta(1 - p)p'$. Unlike Carpin and Thayer [2022], who backup $F$-values based on the best-case, we backpropagate the $F$-values up the tree similar to $Q$-values:

$$F(b, a) = F(b, a) + \frac{p - F(b, a)}{N(b, a)} \quad (5)$$

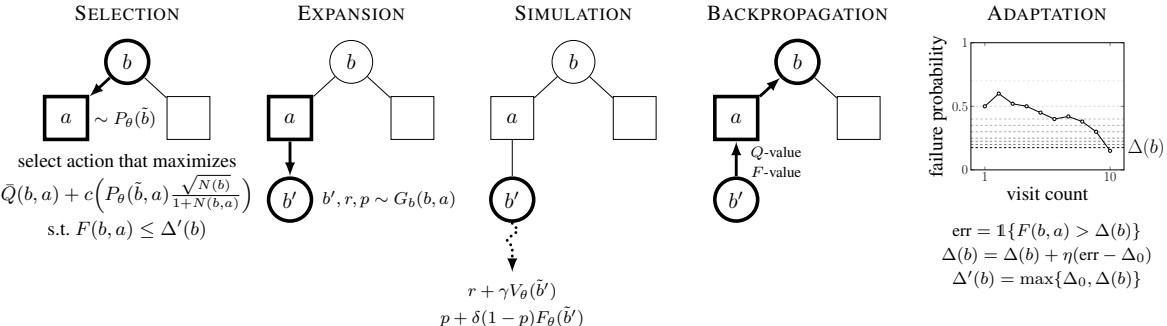

Figure 2: *ConstrainedZero* online Monte Carlo tree search with failure threshold adaptation ($\Delta$-MCTS).

---

**Algorithm 1** $\Delta$-MCTS adaptation.

1: **function** ADAPTATION($\Delta, b, a$)
2:     $l(b) \leftarrow \min_{a' \in A(b)} F(b, a')$         ▷ update bounds
3:     $u(b) \leftarrow \max_{a' \in A(b)} F(b, a')$
4:     err $\leftarrow \mathbb{1}\{F(b, a) > \Delta(b)\}$
5:     $\Delta(b) \leftarrow \text{clip}\big(\Delta(b) + \eta(\text{err} - \Delta_0), l(b), u(b)\big)$

---

which is a running mean estimate where $F(b, a)$ is initialized using the initialization function $F_0(b, a)$ (noting the $F_0$ subscript: which could either be zero, the immediate failure probability $F_b(b, a)$, or the bootstrapped value by taking action $a$ to get a new belief $b'$ and computing $p$ based on the $p' = F_\theta(\tilde{b}')$ estimate).

Using the estimate $F(b, a)$, a simple way to select actions that do not violate the safety constraint set by $\Delta$ would be to use the PUCT algorithm [Silver *et al.*, 2018] with a hard constraint on safety of only choosing actions such that $F(b, a) \le \Delta$ is satisfied. However, if the failure probability threshold $\Delta$ is too conservative, the action-selection process may fail to find *any* action that satisfies the constraint. Therefore, $\Delta$-MCTS tracks an estimate of the threshold $\Delta(b)$ for each belief node and updates it using *adaptive conformal inference* (ACI) [Gibbs and Candes, 2021]. ACI is a statistical method that provides valid prediction intervals without assumptions on how the time-series data was generated. The adaptive threshold is initialized to the target tolerance $\Delta(b) = \Delta_0$ where $\Delta_0 = \Delta$. Each time the $F$-value is updated (either by eq. (5) or initialization), the ADAPTATION procedure is called to update the current acceptable safety threshold.

In adaptation, the error term of err $= \mathbb{1}\{F(b, a) > \Delta(b)\}$ indicates when to widen or restrict the estimated threshold $\Delta(b)$ based on whether the failure probability estimate of the most recently explored belief-action node $F(b, a)$ is above or below the current threshold. The estimated threshold is updated according to

$$\Delta(b) = \Delta(b) + \eta(\text{err} - \Delta_0) \quad (6)$$

which will widen the threshold if the observed failure probability is outside the threshold (i.e., if the error is one), and will tighten the threshold otherwise.

Intuitively, the update adjusts the threshold of acceptable failure probability $\Delta(b)$ based on recent experience. If the failure probability $F(b, a)$ for a recent action is higher than the current threshold $\Delta(b)$, this indicates a higher risk than expected. Thus, the threshold is increased by $\eta(1 - \Delta_0)$ for $\eta > 0$ to allow for more risk in future actions. Otherwise, if $F(b, a)$ is lower than the threshold, this means actions are safer than expected and the threshold is decreased by $\eta \Delta_0$ (favoring a more reactive increase than decrease of the threshold). Notably, Gibbs and Candes [2021] prove that $\Delta(b)$ converges exactly to the desired target over time.

We clip the final threshold to the lower and upper bounds of the observed failure probability for a given belief $b$ to restrict the change in $\Delta(b)$ and, more importantly, to guarantee that at least one action is available for selection (line 5, algorithm 1).

The resulting criterion selects actions that satisfy the adaptive constraint of $F(b, a) \le \Delta'(b)$ where the selection threshold $\Delta'(b) = \max\{\Delta_0, \Delta(b)\}$ upper bounds the failure probability. Together, the $\Delta$-MCTS exploration policy becomes:

$$\pi_{\text{explore}}(b) = \arg\max_{a \in A(b)} \bar{Q}(b, a) + c\Big(P_\theta(\tilde{b}, a) \frac{\sqrt{N(b)}}{1 + N(b, a)}\Big) \quad (7)$$

$$\text{s.t. } F(b, a) \le \Delta'(b) \quad (8)$$

termed the *chance-constrained PUCT* criterion (CC-PUCT). The constraint in eq. (8) is also used to select root actions.

The benefit of CC-PUCT is that when our explored samples satisfy the constraint $\Delta'(b)$ (defined over the belief rather than both belief and action) we may explore new actions from this belief which are both safe and have the potential for higher reward. The key idea is that actions are chosen based on the balance between safety and utility; ensuring that we do not over-prioritize safety at the expense of potential rewards, while not exploiting rewards without regarding the risk.

## 4 Experiments

For a fair comparison, ConstrainedZero was evaluated against BetaZero using the same network and MCTS parameters. BetaZero uses a scalarized reward function to penalize failures, while ConstrainedZero omits the penalty and plans using the adaptive safety constraint instead. The BetaZero reward takes the form $\bar{R}_b(b, a) = R_b(b, a) - \lambda C(b, a)$ with a cost $C$ scaled by $\lambda$. Three safety-critical CC-POMDPs were evaluated. The first is the *LightDark* POMDP, a standard benchmark localization task [Platt Jr. *et al.*, 2010]. The next CC-POMDP is

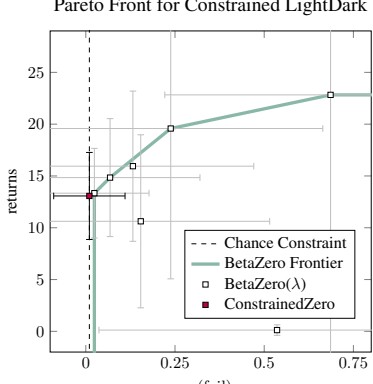

Figure 3: BetaZero($\lambda$) comparison.

| | LightDark $\Delta_0 = 0.01$ | | Collision Avoidance $\Delta_0 = 0.01$ | | Spillpoint CCS $\Delta_0 = 0.05$ | |
|---|---|---|---|---|---|---|
| | $p(\text{fail}) \downarrow$ | returns $\uparrow$ | $p(\text{fail}) \downarrow$ | returns $\uparrow$ | $p(\text{fail}) \downarrow$ | returns $\uparrow$ |
| ConstrainedZero | $\mathbf{0.01}_{\pm 0.01}$ | $\mathbf{13.07}_{\pm 0.42}$ | $\mathbf{0.00}_{\pm 0.00}$ | $\mathbf{-0.74}_{\pm 0.03}$ | $\mathbf{0.05}_{\pm 0.02}$ | $\mathbf{2.62}_{\pm 0.12}$ |
| No Adaptation* | $0.66_{\pm 0.05}$ | $27.47_{\pm 3.90}$ | $0.03_{\pm 0.02}$ | $-1.00_{\pm 0.00}$ | $0.69_{\pm 0.04}$ | $6.18_{\pm 0.36}$ |
| $\Delta$-MCTS (no $f_\theta$)$^\dagger$ | $0.01_{\pm 0.01}$ | $1.86_{\pm 0.20}$ | $0.32_{\pm 0.05}$ | $0.00_{\pm 0.00}$ | $1.00_{\pm 0.00}$ | $6.87_{\pm 0.50}$ |
| Raw Policy $P_\theta$ | $0.01_{\pm 0.01}$ | $12.88_{\pm 0.46}$ | $0.00_{\pm 0.00}$ | $-0.86_{\pm 0.02}$ | $0.06_{\pm 0.02}$ | $2.45_{\pm 0.11}$ |
| Raw Value$^\ddagger$ $V_\theta$ | $0.72_{\pm 0.05}$ | $28.00_{\pm 4.51}$ | $0.16_{\pm 0.04}$ | $-0.20_{\pm 0.04}$ | $0.38_{\pm 0.05}$ | $4.27_{\pm 0.30}$ |
| Raw Failure$^\ddagger$ $F_\theta$ | $0.80_{\pm 0.04}$ | $0.05_{\pm 0.04}$ | $0.00_{\pm 0.00}$ | $-1.62_{\pm 0.08}$ | $0.00_{\pm 0.00}$ | $0.00_{\pm 0.00}$ |

All results report the mean $\pm$ standard error over 100 seeds, evaluated using the argmax of the tree policy.
* Trained with the same parameters as ConstrainedZero without adaptation, i.e., only a hard constraint on $\Delta_0$.
$^\dagger$ $\Delta$-MCTS without the neural network for the value or failure probability and a random policy for CC-PUCT.
$^\ddagger$ One-step look-ahead over all actions using only the value or failure probability network head with 5 obs. per action.

Table 1: ConstrainedZero results. Bold indicates the best results within the $\Delta_0$ threshold.

the *aircraft collision avoidance* problem (CAS), modeled after ACAS X [Kochenderfer *et al.*, 2012]. In the CAS problem, the ownship aircraft attempts to avoid a near mid-air collision (NMAC) with an intruding aircraft while minimizing the alert and reversal rates. Lastly, we study safe *carbon capture and storage* (CCS) [Corso *et al.*, 2022]. A challenge of CCS is safely injecting $CO_2$ into the subsurface while mitigating risk of leakage and earthquakes.

### 4.1 Empirical Results

Figure 3 compares ConstrainedZero against BetaZero, where BetaZero uses different values of the penalty $\lambda$. The penalties were swept between $-10$ and $-1000$ with $-100$ being the standard for the LightDark POMDP (proportional to the goal reward of 100). A target safety level of $\Delta_0 = 0.01$ was chosen for ConstrainedZero. ConstrainedZero exceeds the BetaZero Pareto curve and achieves the target level of safety with a failure probability of $0.01 \pm 0.01$ computed over 100 episodes. BetaZero still achieves good performance but at the cost of sweeping the penalty values without explicitly defining a safety threshold to satisfy.

Shown in table 1, an ablation study is conducted for ConstrainedZero. Most notably, the adaptation procedure is crucial to enable the algorithm to properly balance safety and utility during planning (also shown in fig. 4a–4b). When comparing $\Delta$-MCTS without network approximators against ConstrainedZero, it is clear that offline policy iteration allows for better online planning. ConstrainedZero consistently achieves the highest return within the satisfied safety target.

Compared to BetaZero, fig. 4a and fig. 4b highlight that ConstrainedZero satisfies the safety constraint earlier during policy iteration, while simultaneously maximizing returns (shown for the CAS problem). The policy trained without adaptation learns to maximize returns but fails to satisfy the safety constraint. This is because without adaptation, the algorithm will attempt to satisfy a fixed constraint, not taking into account the outcomes of its actions. With adaptation, ConstrainedZero adjusts the constraint in response to feedback from the environment, resulting in the algorithm becoming more capable at optimizing its performance within the bounds of the adaptive constraint. This demonstrates the importance of adaptation, as a fixed constraint may be too conservative or too risky, leading to suboptimal decision-making.

## 5 Conclusions

This work introduces *ConstrainedZero*, an extension of the BetaZero POMDP planning algorithm to CC-POMDPs. Along with neural network estimates of the value function and action-selection policy, we include a network head that estimates the failure probability given a belief. By framing the safe planning problem as a CC-POMDP, we select a target level of safety to optimize towards, instead of tuning the reward function to balance safety and utility. We develop an extension to MCTS that includes an *adaptation* stage that adjusts the target level of safety during planning using adaptive conformal inference. The resulting $\Delta$-MCTS algorithm modifies MCTS for CC-POMDPs and addresses the issue of overfitting to failure predictions.

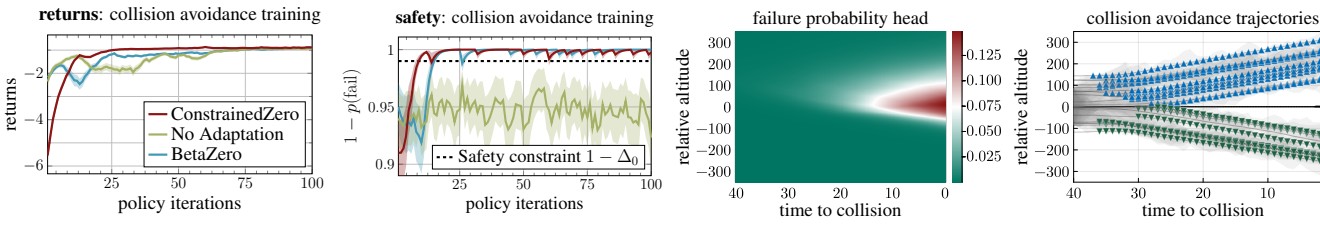

(a) Returns from policy iteration. (b) Safety from policy iteration. (c) Higher $p(\text{fail})$ near an NMAC. (d) Climb (blue), descend (green).

Figure 4: Results for the collision avoidance CC-POMDP. Figure 4d matches the "notch" behavior from Kochenderfer *et al.* [2012].

## Acknowledgments

This research is funded by OMV and J.F. thanks the Karlsruhe House of Young Scientists (KHYS) for travel grant funding.

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
