# OpenReview forum: "Chance-Constrained POMDP Planning with Learned Neural Network Surrogates"
_ijcai.org/IJCAI/2024/Workshop/TIDMwFM — IJCAI TIDMwFM 2024 Oral_

### Official Review · Reviewer_E5WN · 2024-06-19

**Rating:** 8
**Confidence:** 4

**Review:**

This paper presents a significant advancement in the field of safe planning in uncertain environments. The main contribution is the ConstrainedZero policy iteration algorithm, which combines offline neural network training and online Monte Carlo Tree Search (MCTS) with adaptive conformal inference to solve chance-constrained POMDPs (CC-POMDPs). This approach ensures that agents can achieve a target level of safety without compromising on the optimization of rewards.

One of the key strengths of this paper is its novel integration of neural network surrogates to estimate failure probabilities alongside value functions and policies. This addition enables the ConstrainedZero algorithm to prioritize safe actions effectively, a crucial capability for safety-critical applications such as aircraft collision avoidance and CO2 storage safety. The use of adaptive conformal inference to update safety thresholds dynamically during planning further enhances the algorithm's robustness and adaptability.

Empirical results demonstrate that ConstrainedZero outperforms existing methods like BetaZero, particularly in maintaining safety while optimizing performance. The comparison and ablation studies provide strong evidence of the algorithm's efficacy and the importance of its adaptive safety constraints.

Overall, the paper makes a significant contribution to the theme of "Trustworthy Interactive Decision-Making with Foundation Models Workshop" by presenting a method that enhances the reliability and safety of decision-making processes in uncertain environments. The introduction of adaptive safety constraints and the effective use of neural network surrogates mark this work as both innovative and highly relevant.

---

### Official Review · Reviewer_jXfi · 2024-06-21

**Rating:** 9
**Confidence:** 4

**Review:**

This study introduces ConstrainedZero, an algorithm for solving chance-constrained partially observable Markov decision processes (CC-POMDPs). Specifically, an adaptive Monte Carlo Tree Search algorithm is proposed to estimate failure probabilities along with Q-values and adjust the failure probability threshold using adaptive conformal inference. Moreover, A policy iteration algorithm that extends BetaZero for CC-POMDPs, including an additional neural network head, is leveraged to estimate failure probability given a belief.

Strengths:

1. Novel approach: The paper introduces a new method for solving CC-POMDPs, addressing the challenge of balancing safety constraints with reward maximization.

2. Adaptive safety constraints: The algorithm's ability to adapt safety thresholds during planning is innovative and addresses the issue of overfitting to failure predictions.

3. Empirical results: The approach is tested on three relevant safety-critical domains, showing improved performance compared to baseline methods.

4. Theoretical grounding: The use of adaptive conformal inference provides a solid theoretical foundation for the threshold adaptation mechanism.

5. Practical applicability: The method shows promise for real-world safety-critical applications such as aircraft collision avoidance.

Weaknesses:

1. Complexity: The approach involves multiple components and may be challenging to implement and tune for new problems.

2. Scalability: While the method improves over baselines, it needs to be clarified how well it would scale to very large state and action spaces.

3. Hyperparameter sensitivity: The study does not discuss the method's sensitivity to various hyperparameters, such as the learning rate for threshold adaptation.

---

### Decision · Program_Chairs · 2024-06-24

Accept (Oral)